# UNSUPERVISED CROSS-LINGUAL REPRESENTATION LEARNING FOR SPEECH RECOGNITION

## ABSTRACT

This paper presents XLSR which learns cross-lingual speech representations by pretraining a single model from the raw waveform of speech in multiple languages. We build on wav2vec 2.0 which is trained by solving a contrastive task over masked latent speech representations and jointly learns a quantization of the latents shared across languages. The resulting model is fine-tuned on labeled data and experiments show that cross-lingual pretraining significantly outperforms monolingual pretraining. On the CommonVoice benchmark, XLSR shows a relative phoneme error rate reduction of 72% compared to the best known results. On BABEL, our approach improves word error rate by 16% relative compared to a comparable system. Our approach enables a single multilingual speech recognition model which is competitive to strong individual models. Analysis shows that the latent discrete speech representations are shared across languages with increased sharing for related languages.

## 1 INTRODUCTION

Cross-lingual learning aims to build models which leverage data from other languages to improve performance. This has been a long standing interest in the speech community (Byrne et al., 2000; Le & Besacier, 2009; Ghoshal et al., 2013; Huang et al., 2013; Gales et al., 2017; Cho et al., 2018; Seki et al., 2018) which includes systems able to transcribe multiple languages (Burget et al., 2010; Bourlard et al., 2011; Heigold et al., 2013; Toshniwal et al., 2018; Kannan et al., 2019). However, the vast majority of work in speech processing has focused on supervised cross-lingual training which requires labeled data in multiple languages. Transcribed speech is often much scarcer than unlabeled speech and requires non-trivial human annotation.

Unsupervised representation learning, or pretraining, does not require labeled data and has received a lot of recent attention in computer vision (Tian et al., 2019; He et al., 2019; Chen et al., 2020) after much success in natural language processing (Peters et al., 2018; Devlin et al., 2018). For the latter, cross-lingual pretraining has been shown to be very effective, particularly, for low resource languages (Lample & Conneau, 2019; Conneau et al., 2019). In speech processing, most work in this area has focused on monolingual unsupervised representation learning (van den Oord et al., 2018; Chung & Glass, 2018; Schneider et al., 2019; Chung et al., 2019; Baevski et al., 2020b; Harwath et al., 2020; Jiang et al., 2019; Tjandra et al., 2019; Eloff et al., 2019; Baevski et al., 2020a).

In this paper, we focus on the cross-lingual setting by learning representations on unlabeled data that generalize across languages. We build on the pretraining approach of Baevski et al. (2020c) which jointly learns contextualized speech representations as well as a discrete vocabulary of latent speech representations. The latter serves to effectively train the model with a contrastive loss (§ 2) and the discrete speech representations are shared across languages (Figure 1). Different to recent work on unsupervised cross-lingual pretraining, we fine-tune the Transformer part of the model instead of freezing all pretrained representations (Rivière et al., 2020) or feeding them to a separate downstream model (Kawakami et al., 2020). We extend the work of Rivière et al. (2020) by pretraining on multiple languages instead of just English and we experiment on top of a stronger baseline.

We evaluate XLSR on 14 languages of the BABEL benchmark (Gales et al., 2014) which is conversational telephone data and ten languages of CommonVoice (Ardila et al., 2019), a corpus of read speech (§ 3). Multilingual pretraining outperforms monolingual pretraining in most cases, except for resource rich languages and we show that increased model capacity significantly closes the gap. We

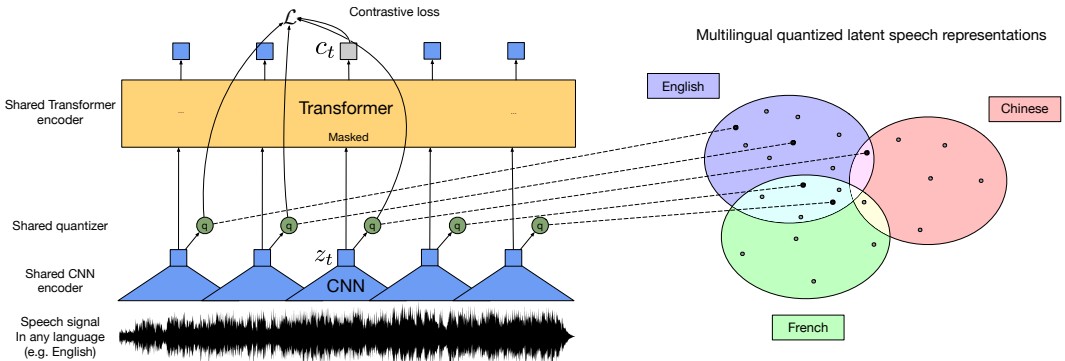

Figure 1: **The XLSR approach.** A shared quantization module over feature encoder representations produces multilingual quantized speech units whose embeddings are then used as targets for a Transformer trained by contrastive learning. The model learns to share discrete tokens across languages, creating bridges across languages. Our approach is inspired by Devlin et al. (2018); Lample & Conneau (2019) and builds on top of wav2vec 2.0 (Baevski et al., 2020c). It requires only raw unlabeled speech audio in multiple languages.

also demonstrate that XLSR representations can be fine-tuned simultaneously on multiple languages to obtain a multilingual speech recognition system whose performance is competitive to fine-tuning a separate model on each language § 4).

## 2 APPROACH

Unsupervised cross-lingual representation learning has shown great success by pretraining Transformers (Vaswani et al., 2017) with multilingual masked language models (Devlin et al., 2018; Lample & Conneau, 2019). In this work, we learn cross-lingual speech representations by extending wav2vec 2.0 (Baevski et al., 2020c) to the cross-lingual setting. Our approach learns a single set of quantized latent speech representations which are shared across languages. Next, we outline the architecture (§ 2.1), training (§ 2.2) and adaptations for cross-lingual training.

### 2.1 ARCHITECTURE

We follow the design choices described in Baevski et al. (2020c). The model contains a convolutional feature encoder $f : \mathcal{X} \mapsto \mathcal{Z}$ to map raw audio $\mathcal{X}$ to latent speech representations $\mathbf{z}_1, \ldots, \mathbf{z}_T$ which are fed to a Transformer network $g : \mathcal{Z} \mapsto \mathcal{C}$ to output context representations $\mathbf{c}_1, \ldots, \mathbf{c}_T$ (Devlin et al., 2018; Baevski et al., 2020b;a). For the purpose of training the model, feature encoder representations are discretized to $\mathbf{q}_1, \ldots, \mathbf{q}_T$ with a quantization module $\mathcal{Z} \mapsto \mathcal{Q}$ to represent the targets in the self-supervised learning objective (Figure 1, § 2.2).

The quantization is based on product quantization (Jegou et al., 2011; Baevski et al., 2020b) by choosing quantized representations from $G = 2$ codebooks with $V = 320$ entries each. The result is concatenated to obtain $\mathbf{q}$. A Gumbel softmax enables choosing discrete codebook entries in a fully differentiable way (Jang et al., 2016). Each $\mathbf{z}_t$ represents about 25ms of audio strided by 20ms, the context network architecture follows BERT (Vaswani et al., 2017; Devlin et al., 2018) except for relative positional embeddings Mohamed et al. (2019); Baevski et al. (2020a).

### 2.2 TRAINING

The model is trained by solving a contrastive task over masked feature encoder outputs. For masking, we sample $p = 0.065$ of all time steps to be starting indices and mask the subsequent $M = 10$ time steps. The objective requires identifying the true quantized latent $\tilde{\mathbf{q}}$ for a masked time-step within a set of $K = 100$ distractors $\mathbf{Q}_t$ sampled from other masked time steps: $-\log \frac{\exp(sim(\mathbf{c}_t, \mathbf{q}_t))}{\sum_{\tilde{\mathbf{q}} \sim \mathbf{Q}_t} \exp(sim(\mathbf{c}_t, \tilde{\mathbf{q}}))}$ where $\mathbf{c}_t$ is the output of the transformer, and $sim(\mathbf{a}, \mathbf{b})$ denotes cosine similarity.

This is augmented by a codebook diversity penalty to encourage the model to use all codebook entries (Dieleman et al., 2018). We maximize the entropy of the averaged softmax distribution over the codebook entries for each group $\bar{p}_g$ across a batch of utterances: $\frac{1}{GV} \sum_{g=1}^{G} -H(\bar{p}_g) = \frac{1}{GV} \sum_{g=1}^{G} \sum_{v=1}^{V} \bar{p}_{g,v} \log \bar{p}_{g,v}$. To stabilize the feature encoder we apply an L2 penalty over the outputs of the feature encoder.

When pretraining on $L$ languages, we form multilingual batches (Devlin et al., 2018; Lample & Conneau, 2019) by sampling speech samples from a multinomial distribution $(p_l)_{l=1,\ldots,L}$ where $p_l \sim \left(\frac{n_l}{N}\right)^{\alpha}$, $n_l$ being the number of pretraining hours of language $l$, $N$ the total number of hours, and $\alpha$ the upsampling factor. The parameter $\alpha$ controls the importance given to high-resource versus low-resource languages during pretraining.

## 3 EXPERIMENTAL SETUP

### 3.1 DATASETS

**CommonVoice.**   The CommonVoice dataset[1] is a multilingual corpus of read speech comprising more than two thousand hours of speech data in 38 languages (Ardila et al., 2019). The amount of data per language ranges from three hours for Swedish ("low-resource") to 353 hours for French and 1350 hours for English ("high-resource"). Following Rivière et al. (2020) we consider ten languages: *Spanish (es), French (fr), Italian (it), Kyrgyz (ky), Dutch (du), Russian (ru), Swedish (sv), Turkish (tr), Tatar (tt) and Chinese (zh)*; as well as English (en) for pretraining. We use the November 2019 release for training models, and for fine-tuning we use the evaluation splits of Rivière et al. (2020) which include one hour labeled data for training, 20 minutes for validation and one hour for testing. This few-shot evaluation dataset consists of phoneme sequences as output and we report phone error rate (PER) similar to prior work.

**BABEL.**   This dataset[2] is a multilingual corpus of conversational telephone speech from IARPA, which includes Asian and African languages (Gales et al., 2014). We adopt the same setup as Cho et al. (2018) and pretrain on ten languages: *Bengali (bn), Cantonese (zh), Georgian (ka), Haitian (ht), Kurmanji (ku), Pashto (ps), Tamil (ta), Turkish (tr), Tokpisin (tp), Vietnamese (vi)*. We evaluate cross-lingual transfer on four other languages (models are not pretrained on these languages): *Assamese (as), Tagalog (tl), Swahili (sw), Lao (lo)*. We train a multilingual model in ten languages and monolingual models in 14 languages. We use the same speech audio for pretraining and fine-tuning, and no unlabeled speech provided by BABEL. We use the dev folder of the BABEL dataset as our test set as "eval" has not been open-sourced, and use 10% of the training set as dev data. We report character error rate (CER). All audio is resampled to 16kHz. For comparison with Inaguma et al. (2019) only, we train 4-gram n-gram language models on CommonCrawl data (Heafield et al., 2013; Wenzek et al., 2019) for Assamese (140MiB of text data), Swahili (2GiB), Tamil (4.8GiB) and Lao (763MiB).

### 3.2 TRAINING DETAILS

**Pretraining**   Models are implemented in fairseq (Ott et al., 2019). We evaluate two architectures with the same feature encoder (§ 2.1) but different Transformer settings: Base with 12 blocks, model dimension 768, inner dimension (FFN) 3072 and 8 attention heads; and Large with 24 blocks, model dimension 1024, inner dimension 4096 and 16 attention heads; both use dropout 0.1. For Base, we crop 250k samples, or 15.6sec of audio, and pack up to 1.4m samples on each GPU. For Large, we crop 320k samples and put up to 1.2m samples on a GPU. Batches are sampled using a factor $\alpha \in \{0.5, 1\}$. We use 16 GPUs for small datasets (typically monolingual) and 64 GPUs for large datasets (typically multilingual), and use Adam (Kingma & Ba, 2015) where the learning rate is warmed up for the first 10% of updates to a peak of 1e-5 (Base) or 1e-3 (Large), and then linearly decayed over a total of 250k updates.

**Fine-tuning.**   To fine-tune the model we add a classifier representing the output vocabulary of the respective downstream task on top of the model and train on the labeled data with a Connectionist

---

[1] https://voice.mozilla.org/en/languages

[2] https://catalog.ldc.upenn.edu/byyear, includes LDC2018S07, LDC2018S13, LDC2018S02, LDC2017S03, LDC2017S22, LDC2017S08, LDC2017S05, LDC2017S13, LDC2017S01, LDC2017S19, LDC2016S06, LDC2016S08, LDC2016S02, LDC2016S12, LDC2016S09, LDC2016S13, LDC2016S10.

| Model | D | #pt | #ft | es | fr | it | ky | nl | ru | sv | tr | tt | zh | Avg |
|---|---|---|---|---|---|---|---|---|---|---|---|---|---|---|
| Number of pretraining hours per language | | | | 168h | 353h | 90h | 17h | 29h | 55h | 3h | 11h | 17h | 50h | 793h |
| Number of fine-tuning hours per language | | | | 1h | 1h | 1h | 1h | 1h | 1h | 1h | 1h | 1h | 1h | 10h |
| *Baselines from previous work* | | | | | | | | | | | | | | |
| m-CPC[†] (Rivière et al., 2020) | $LS_{100h}$ | 10 | 1 | 38.7 | 49.3 | 42.1 | 40.7 | 44.4 | 45.2 | 48.8 | 49.7 | 44.0 | 55.5 | 45.8 |
| m-CPC[†] (Rivière et al., 2020) | $LS_{360h}$ | 10 | 1 | 38.0 | 47.1 | 40.5 | 41.2 | 42.5 | 43.7 | 47.5 | 47.3 | 42.0 | 55.0 | 44.5 |
| Fer et al.[†] (Fer et al., 2017) | $BBL_{all}$ | 10 | 1 | 36.6 | 48.3 | 39.0 | 38.7 | 47.9 | 45.2 | 52.6 | 43.4 | 42.5 | 54.3 | 44.9 |
| *Our monolingual models* | | | | | | | | | | | | | | |
| XLSR-English | $CV_{en}$ | 1 | 1 | 13.7 | 20.0 | 19.1 | 13.2 | 19.4 | 18.6 | 21.1 | 15.5 | 11.5 | 27.1 | 17.9 |
| XLSR-Monolingual | $CV_{mo}$ | 1 | 1 | **6.8** | **10.4** | **10.9** | 29.6 | 37.4 | 11.6 | 63.6 | 44.0 | 21.4 | 31.4 | 26.7 |
| *Our multilingual models* | | | | | | | | | | | | | | |
| XLSR-10 (unbalanced) | $CV_{all}$ | 10 | 1 | 9.7 | 13.6 | 15.2 | 11.1 | 18.1 | 13.7 | 21.4 | 14.2 | 9.7 | 25.8 | 15.3 |
| XLSR-10 | $CV_{all}$ | 10 | 1 | 9.4 | 14.2 | 14.1 | 8.4 | 16.1 | 11.0 | 20.7 | 11.2 | 7.6 | 24.0 | 13.6 |
| XLSR-10 (separate vocab) | $CV_{all}$ | 10 | 10 | 10.0 | 13.8 | 14.0 | 8.8 | 16.5 | 11.6 | 21.4 | 12.0 | 8.7 | 24.5 | 14.1 |
| XLSR-10 (shared vocab) | $CV_{all}$ | 10 | 10 | 9.4 | 13.4 | 13.8 | 8.6 | 16.3 | 11.2 | 21.0 | 11.7 | 8.3 | 24.5 | 13.8 |
| *Our multilingual models (Large)* | | | | | | | | | | | | | | |
| XLSR-10 | $CV_{all}$ | 10 | 1 | 7.9 | 12.6 | 11.7 | **7.0** | 14.0 | **9.3** | **20.6** | **9.7** | 7.2 | 22.8 | 12.3 |
| XLSR-10 (separate vocab) | $CV_{all}$ | 10 | 10 | 8.1 | 12.1 | 11.9 | 7.1 | 13.9 | 9.8 | 21.0 | 10.4 | 7.6 | **22.3** | 12.4 |
| XLSR-10 (shared vocab) | $CV_{all}$ | 10 | 10 | 7.7 | 12.2 | 11.6 | **7.0** | **13.8** | **9.3** | 20.8 | 10.1 | 7.3 | **22.3** | **12.2** |

Table 1: **CommonVoice phoneme error rate (PER).** Models are pretrained on either one language (pt = 1) or 10 languages (pt = 10); and fine-tuned on each language (ft = 1) or all languages (ft = 10). D indicates the pretraining data, LS for English LibriSpeech (100h or 360h), $BBL_{all}$ for BABEL (1070h), $CV_{En}$ for English CommonVoice (1350h), $CV_{mo}$ for monolingual (see number of pretraining hours per language) and $CV_{all}$ for multilingual (1350h). Languages can be high-resource (es, fr, it) or low-resource (e.g. ky, sv, tr, tt). Baseline results [†] are from Rivière et al. (2020).

Temporal Classification (CTC) loss (Graves et al., 2006; Baevski et al., 2020a). Weights of the feature encoder are not updated at fine-tuning time. We determine the best learning rates setting in [2e-5, 6e-5] based on dev set error rate. The learning rate schedule has three phases: warm up for the first 10% of updates, keep constant for 40% and then linearly decay for the remainder. For CommonVoice we fine-tune for 20k updates and on BABEL for 50k updates on 2 GPUs for the Base model and 4 GPUs for the Large model.

## 3.3 PRETRAINED MODELS

We use the Base architecture unless otherwise stated. For CommonVoice, we pretrain an English model on 1350h, and ten monolingual models on each pretraining set. For comparison with the English model, we train Base and Large multilingual models on 1350h of data: 793h of speech audio from the 10 evaluation languages plus 557h of English audio. We upsample low-resource languages with $\alpha = 0.5$ and train a model with $\alpha = 1$ for comparison (unbalanced). For multilingual fine-tuning, we either separate or share phoneme vocabularies across languages.

For BABEL, we train a monolingual model on each of the 14 languages, as well as a Base and Large multilingual model on a total of 650 hours of speech audio in ten languages. Since the amount of data in each language is more balanced than for CommonVoice, we use $\alpha = 1$. The same speech audio is used for pretraining and fine-tuning and we use separate character sets for multilingual fine-tuning.

## 4 RESULTS

In our experiments, we first show that our approach is very effective for learning generic cross-lingual representations in an unsupervised way. Pretraining a single model on multiple languages significantly outperforms the previous state of the art on CommonVoice, as well as our own monolingual models. Second, we demonstrate the positive impact of cross-lingual transfer on low-resource languages and provide a better understanding of the trade-off between high-resource and low-resource languages. Third, by fine-tuning a multilingual model on many languages at once, we show that we can obtain a single model for all languages with strong performance. Finally, we analyse the impact of language similarity on cross-lingual transfer, and show that, to some extent, our multilingual pretrained model implicitly learns to cluster related languages.

### 4.1 Effectiveness of unsupervised cross-lingual representation learning

In what follows, we compare XLSR to several baselines and show that unsupervised cross-lingual representation learning is very effective. We provide a comprehensive analysis of the impact of different pretraining methods on automatic speech recognition in Table 1, 2 and 3.

#### 4.1.1 Multilingual outperforms monolingual pretraining and prior art

We first compare monolingual (XLSR-Monolingual) to multilingual (XLSR-10) pretrained models (Base) fine-tuned individually on each language (ft=1). On CommonVoice, XLSR-10 obtains 13.6 PER on average (*Avg*), a relative PER reduction of 49% compared to XLSR-Monolingual (Table 1). On BABEL, XLSR-10 improves over XLSR-Monolingual by 18% relative CER (Table 2) and by more over supervised training (Training from scratch).[3] Pretraining on multiple languages results in cross-lingual transfer and better speech representations.

Compared to prior work, XLSR-10 Large reduces PER by 72% relative to m-CPC (Rivière et al., 2020) on CommonVoice (Table 1). For BABEL, we compare to the supervised multilingual model of Cho et al. (2018) whose setup we adopted: XLSR-10 Large reduces CER by 38% relative to multi-BLSTMP+VGG (Table 2). The other most comparable work is Inaguma et al. (2019) and Table 4 shows a relative word error reduction of 16% compared to their monolingual baseline (BLSTM-HMM) which outperforms their own supervised multilingual model (Multi-10).

#### 4.1.2 Multilingual pretraining outperforms English-only training

To isolate the impact of multilingual training versus simply training on more data, we pretrain an English-only CommonVoice model (XLSR-English) on the same amount of data as the multilingual model (1350h) and compare the two. Table 1 shows that on average, XLSR-English significantly improves over the monolingual models (average PER of 26.7 vs. 17.9 PER) but multilingual pretraining performs even better at 13.6 PER, a 24% relative PER reduction over XLSR-English. This shows that adding more training data is not the only reason for the improved accuracy: the similarity between the languages used in pretraining and fine-tuning also plays an important role.

#### 4.1.3 Learned representations transfer well to unseen languages

To better assess the cross-lingual transfer of the learned representations, we evaluate the XLSR-10 BABEL model on four languages not seen during pretraining. We fine-tune this model on each language, and compare it to monolingual models pretrained specifically on these languages. Table 3 shows that a multilingual model not pretrained on any data from the four languages, still outperforms XLSR-Monolingual, reducing average CER from 29 to 22.8 which compares to results from previous work of 36.8 CER (Cho et al., 2018). This further suggests that the learned representations capture generic features of the speech signal which transfer to many languages.

### 4.2 Understanding cross-lingual transfer learning

In this section, we examine several properties of unsupervised cross-lingual representation learning for speech recognition. We show that it is particularly effective on low-resource languages, then describe the transfer-interference trade-off which benefits low resource languages but hurts high resource languages. Finally, we show that adding capacity is important for multilingual pretraining.

#### 4.2.1 Cross-lingual transfer learning improves low-resource language understanding

Unsupervised cross-lingual representation learning and cross-lingual transfer are particularly effective on low-resource languages. On CommonVoice, the separation between high-resource and low-resource languages is more salient than for BABEL. We distinguish between low-resource and high-resource based on the amount of available unlabeled speech data. For example, French and Spanish have 353h and 168h and are thus high-resource, while Swedish and Turkish have 3h and 11h and are low-resource. Monolingual models perform poorly on low-resource languages but this is

---

[3]We did not tune this setting extensively and will tune it further in the next version of the paper.

| Model | #pt | #ft | bn | zh | ka | ht | ku | ps | ta | tr | tp | vi | Avg |
|---|---|---|---|---|---|---|---|---|---|---|---|---|---|
| Number of pretraining hours per language | | | 56h | 130h | 46h | 61h | 38h | 71h | 63h | 70h | 36h | 79h | 650h |
| Number of fine-tuning hours per language | | | 56h | 130h | 46h | 61h | 38h | 71h | 63h | 70h | 36h | 79h | 650h |
| *Baselines from previous work* | | | | | | | | | | | | | |
| Mono-BLSTMP (Cho et al., 2018) | 10 | 1 | 43.4 | 37.4 | 35.4 | 39.7 | 55.0 | 37.3 | 55.3 | 50.3 | 32.7 | 54.3 | 44.1 |
| Multi-BLSTMP (Cho et al., 2018) | 10 | 1 | 42.9 | 36.3 | 38.9 | 38.5 | 52.1 | 39.0 | 48.5 | 36.4 | 31.7 | 41.0 | 40.5 |
| + VGG (Cho et al., 2018) | 10 | 1 | 39.6 | 34.3 | 36.0 | 34.5 | 49.9 | 34.7 | 45.5 | 28.7 | 33.7 | 37.4 | 37.4 |
| *Our monolingual models* | | | | | | | | | | | | | |
| Training from scratch | 1 | 1 | 47.6 | 42.7 | 45.0 | 45.0 | 58.4 | 43.2 | 55.7 | 44.6 | 45.2 | 43.6 | 47.1 |
| XLSR-Monolingual | 1 | 1 | 31.8 | 28.0 | 30.5 | 27.9 | 46.9 | 25.5 | 36.0 | 26.1 | 26.8 | 25.2 | 30.5 |
| *Our multilingual models* | | | | | | | | | | | | | |
| XLSR-10 | 10 | 1 | 26.6 | 24.7 | 21.8 | 23.2 | 38.2 | 22.6 | 30.5 | 22.3 | 17.3 | 21.7 | 24.9 |
| XLSR-10 (separate vocab) | 10 | 10 | 29.5 | 29.1 | 25.9 | 26.5 | 40.4 | 25.8 | 33.4 | 24.6 | 19.3 | 24.3 | 27.9 |
| *Our multilingual models (Large)* | | | | | | | | | | | | | |
| XLSR-10 | 10 | 1 | **25.1** | **23.4** | **19.7** | **21.1** | **36.8** | 21.6 | **28.6** | **19.8** | 16.1 | **19.9** | **23.2** |
| XLSR-10 (separate vocab) | 10 | 10 | 25.8 | 25.0 | 20.7 | 22.0 | 37.2 | **21.2** | 28.9 | 19.9 | **15.9** | 20.7 | 23.7 |

Table 2: **BABEL results using character error rate (CER) on in-pretraining languages.** Baseline results are from Cho et al. (2018) and use the same amount of data as our multilingual models.

where cross-lingual transfer is most effective: XLSR-10 reduces PER over XLSR-Monolingual by a relative 67% on Swedish, 72% on Turkish, 72% on Kyrgyz, and 64% on Tatar.

On BABEL, the amount of monolingual data ranges between 30 hours for Swahili and 130 hours for Cantonese, with a mean of 65h per language. The results (Table 2 and 3) show that the multilingual model outperforms the monolingual model on all languages, but the biggest gains are obtained on the four lowest-resource languages: Georgian (ka), Kurmanji (ku), Tokpisin (tp) and Swahili (sw).

### 4.2.2 THE TRANSFER-INTERFERENCE TRADE-OFF: HIGH-RESOURCE VS. LOW-RESOURCE

Per language results on CommonVoice (Table 1) show *transfer-interference* trade-off (Arivazhagan et al., 2019): for low-resource languages (e.g. ky, nl, sv, tr, tt), multilingual models outperform monolingual models because of positive transfer, however multilingual models perform worse on high-resource languages (es, fr, it), due to *interference*. Data from multiple languages enables better speech representations that transfer to low-resource languages but the model also needs to share its capacity across languages which degrades performance on high-resource languages.

For a given model capacity, the language sampling parameter $\alpha$ (see § 2) controls this trade-off. Table 1 shows that training according to the true language distribution, XLSR-10 (unbalanced) using $\alpha = 1$, performs less well than XLSR-10, where more capacity is allocated to low-resource languages via $\alpha = 0.5$. The sole exception being French, the language with the most data. On average the unbalanced model obtains 15.3 PER while the balanced model obtains 13.6.

### 4.2.3 INCREASING CAPACITY FOR A MULTILINGUAL PRETRAINED MODEL

The interference problem can be alleviated by adding more capacity to the multilingual model (Arivazhagan et al., 2019; Conneau et al., 2019): the gap between multilingual models and monolingual models for high-resource languages can be reduced by increasing model capacity. In this work, we only study the impact of adding more capacity to the multilingual model, by training an XLSR-10 Large model. On CommonVoice, the Large model reduces PER by relative 9.6% compared to Base, reducing average PER from 13.6 to 12.3. There are no gains on very low-resource languages like Swedish but significant gains on Spanish, French and Italian. On BABEL, average CER is reduced by a relative 6.8%. This shows that the multilingual model benefits from more capacity overall, and in particular for high-resource languages.

### 4.3 SUPERVISED MULTILINGUAL FINE-TUNING: ONE MODEL FOR ALL LANGUAGES

When we fine-tune the pretrained model on each language individually, then we end up with a different model for each language. On the other hand, multilingual speech recognition aims to build a single model for all languages that performs as well or better than individual monolingual models.

| Model | #pt | #ft | as | tl | sw | lo | Avg |
|---|---|---|---|---|---|---|---|
| Number of pretraining hours | | | 55h | 76h | 30h | 59h | 220h |
| Number of fine-tuning hours | | | 55h | 76h | 30h | 59h | 220h |
| *Baselines from previous work* | | | | | | | |
| Monolingual (Cho et al., 2018) | 10 | 1 | 45.6 | 43.1 | 33.1 | 42.1 | 41.0 |
| Stage-2 retraining (Cho et al., 2018) | 10 | 1 | 41.3 | 37.9 | 29.1 | 38.7 | 36.8 |
| *Our monolingual models* | | | | | | | |
| Training from scratch | 1 | 1 | 50.2 | 41.7 | 40.8 | 43.5 | 44.1 |
| XLSR-Monolingual | 1 | 1 | 34.8 | 25.4 | 26.8 | 29.1 | 29.0 |
| *Our multilingual models* | | | | | | | |
| XLSR-10 | 10 | 1 | 29.4 | 21.9 | 16.6 | 23.3 | 22.8 |
| XLSR-10 (Large) | 10 | 1 | **27.7** | **19.6** | **14.9** | **21.8** | **21.0** |

Table 3: **BABEL results on out-of-pretraining languages (CER)**. XLSR-10 provides strong representations for languages not seen during pretraining, outperforming monolingual models pretrained specifically on these languages.

| Model | #pt | #ft | as | tl | sw | lo | Avg |
|---|---|---|---|---|---|---|---|
| Number of pretraining hours | | | 55h | 76h | 30h | 59h | 220h |
| Number of fine-tuning hours | | | 55h | 76h | 30h | 59h | 220h |
| *Baselines from previous work* | | | | | | | |
| Multi-10 (Inaguma et al., 2019) | 10 | 1 | 53.6 | 46.2 | 41.6 | 45.9 | 46.8 |
| BLSTM-HMM (Inaguma et al., 2019) | 1 | 1 | 49.1 | 46.3 | 38.3 | 45.7 | 44.9 |
| BRNN (Ragni et al., 2018) | 1 | 1 | - | 40.6 | **35.5** | - | - |
| *Our approach (no LM)* | | | | | | | |
| XLSR-10 (Large) | 10 | 1 | 49.1 | 40.6 | 38.1 | 34.7 | 40.6 |
| *Our approach (4-gram KenLM)* | | | | | | | |
| XLSR-10 (Large) | 10 | 1 | **44.9** | **37.3** | **35.5** | **32.2** | **37.5** |

Table 4: **BABEL results on out-of-pretraining languages using word error rate (WER).** XLSR-10 reduces word error rate by 16.5% compared to previously published results on four of the BABEL languages. We report WER with and without 4-gram KenLM language models.

Next, we investigate fine-tuning a single model on the labeled data of all languages (#ft=10) to obtain a single multilingual model instead of fine-tuning each language separately (#ft=1). Training batches are constructed by sampling audio samples from multiple languages (without upsampling).

For CommonVoice we consider two settings since we use phonemes: separate phoneme vocabularies per languages as well as sharing phonemes across languages. A shared vocabulary reduces the number of modeled phonemes from 474 to 182 compared to separate vocabularies. Table 1 shows that the Base model with monolingual fine-tuning of XLSR-10 obtains 13.6 average PER which compares to 14.1 PER and 13.8 PER for separate and shared vocabulary multilingual fine-tuning respectively. When increasing model capacity (Large), multilingual fine-tuning is competitive to monolingual fine-tuning: 12.3 average PER (ft=1) vs. 12.4 and 12.2 average PER (ft=N) for separate and shared vocabularies. Multilingual fine-tuning of the Large model with a shared vocabulary achieves the best overall performance on CommonVoice.

BABEL provides significantly more labeled data (650h for all languages) compared to CommonVoice (10h for all languages). Performance on BABEL with multilingual fine-tuning of the XLSR-10 Base model significantly decreases from 24.9 to 27.9 average CER compared to monolingual fine-tuning. However, increasing capacity helps to counteract this: XLSR-10 Large achieves 23.7 avgerage CER which is much closer to monolingual fine-tuning of the Large model (23.2 avg. CER). Increasing capacity is particularly important when fine-tuning on large amounts of supervised data from many languages. Multilingual fine-tuning performs competitively to monolingual fine-tuning and enables us to have a single model for many languages.

## 4.4 ON THE ROLE OF LANGUAGE SIMILARITY ON CROSS-LINGUAL TRANSFER

Next, we study the impact of language similarity on cross-lingual transfer and then analyze the multilingual token embedding space where we find that languages are clustered.

### 4.4.1 LOW-RESOURCE LANGUAGES BENEFIT MORE FROM SIMILAR HIGHER-RESOURCE LANGUAGES

We consider Italian as the low-resource language for which we assume only 5h of unlabeled data is available. We pretrain models on the 5h as well as 50h of unlabeled data from several other languages:

| Model | #pt | #ft | it | it | es | de | en | ru | ka | zh |
|---|---|---|---|---|---|---|---|---|---|---|
| Number of pretraining hours | | | 5h (it) | 5h (it) + 50h (<lang>) | | | | | | |
| XLSR-Monolingual | 1 | 1 | 47.6 | 16.8 | 24.3 | 25.4 | 27 | 27.2 | 28.1 | 30.6 |

Table 5: **Impact of language similarity on cross-lingual transfer.** We simulate a low-resource language scenario by using only 5 hours of Italian CommonVoice data and add 50 hours from another language for pretraining. We fine-tune on 1 hour of Italian supervised data.

Italian, Spanish, German, English, Russian, Kabyle and Chinese. Finally, we fine-tune each model on 1h of Italian labeled data. Table 5 shows that adding more unlabeled data helps overall, but adding data from related languages gives the largest improvement, e.g., Spanish. Distant languages, e.g., Kabyle or Chinese are less effective. In order to improve performance on a low-resource language, it is best to add unlabeled data from a closely-related language.

### 4.4.2 ANALYZING THE SHARED DISCRETE SPEECH REPRESENTATIONS

To analyze the shared quantized latent speech representations, or discrete tokens, we train two models: one on 12 languages of CommonVoice and another on 17 languages of BABEL. For each model, we run the quantizer of our model on train and dev speech samples from each language, and compute a frequency vector of the discrete tokens. The resulting frequencies are normalized for each language to obtain vectors of size $V \times G$, the number of discrete latent speech representations. The vectors represent the empirical probability distribution over the shared discrete latents. Next, we construct an affinity matrix between languages by computing the Jensen-Shannon symmetric similarity between vectors. Finally, we cluster languages using K-Means and then perform a PCA with two dimensions.

Figure 2a, 2b and 2c show the visualizations, where colors correspond to the clusters obtained by K-Means. Note, that we perform K-Means before PCA to avoid loss of information, and that PCA may make some points appear closer than they are in original vectors. We see that the model shares more discrete tokens for similar languages, e.g., it groups Basque, Catalan, Spanish and Italian, or English, German and French, or Arabic and Kabyle (see Figure 2a), and Mandarin (zh-CN and zh-TW), although this information is lost in the PCA visualization. Figure 2b shows that the model may also isolate a language, such as Chinese-HongKong (Cantonese), which is not close to any other language because it shares fewer discrete tokens with other languages.

For BABEL (Figure 2c), we also find language groupings such as Bengali/Assamese which belong to the same family, or Zulu and Swahili which both have long vowels. However one could argue that Pashto and Kurmanji should be closer to each other since they are both Iranian languages. The purpose of this analysis is not to recover full language families but to better understand how our model allocates the latent representations across languages. Interestingly, Italian is closer to Spanish, the most effective language in the previous experiment (Table 5). Future work may investigate whether encouraging shared tokens between similar languages could further help cross-lingual transfer.

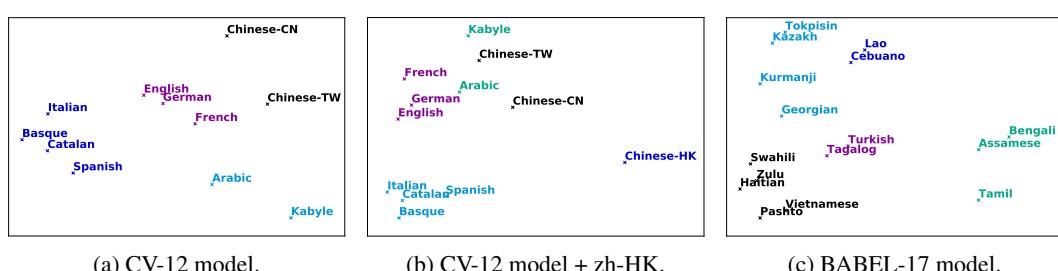

| (a) CV-12 model. | (b) CV-12 model + zh-HK. | (c) BABEL-17 model. |

Figure 2: **Visualization of language similarities learned by the model** Figure (a) visualizes the shared discrete latent speech representations across languages for a model trained on 12 Common-Voice languages (CV-12). Figure (b) shows that adding Chinese-HongKong (zh-HK) shares relatively few latents with other languages. Figure (c) is for a model trained on 17 BABEL languages and illustrates that clusters can correspond to similar languages like Bengali and Assamese.

## 5 CONCLUSION

In this work, we investigated unsupervised cross-lingual speech representations learned from the raw waveform. We show that pretraining on data in multiple languages improves both over monolingual pretraining as well as prior work, with the largest improvements on low-resource languages. Fine-tuning the model on multiple languages at once enables a single multilingual speech recognition model competitive to individually fine-tuned models. Analysis of the discrete latent speech representations reveals that the model shares capacity across languages and particularly so with related languages.

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
