# OpenReview forum: "Unsupervised Cross-lingual Representation Learning for Speech Recognition"
_ICLR.cc/2021/Conference — Reject_

### Official Review · AnonReviewer4 · 2020-10-28
**A very practical and useful study but lacks of novelty**

**Rating:** 6
**Confidence:** 4

**Review:**

This paper proposes an unsupervised cross-lingual speech representation learning algorithm for multi-lingual automatic speech recognition. The main building block is based on an existing study (wav2vec series). The idea of this study is to apply previously (successful) unsupervised speech representation learning scheme to cross-lingual scenario to overcome the data sparsity problem of ASR for low-resource language. In this way, the experiments done in this paper are very meaningful and could shed light on future work on low-resource speech processing. The evaluations are intensive, and the results support most of the main claims well.

Cons:
1. Since the building block is from an existing study: wav2vec 2.0 and there is no improvement to it, this paper lacks of novelty in theoretical part.
2. The experimental design for the investigation on transfer-interference trade-off mainly comes from existing studies (Conneau et al., 2019). No further insight is given. Actually, less analysis is done compared to (Conneau et al., 2019): will the performance on low-resource languages keep increasing if more languages are included.
3. There doesn't exist a clear definition of "similar languages" in section 4.4. Thus the results in section 4.4.2 could be confusing: not sure how these languages are clustered together, because they are overlapping on vowel/consonant space?

Questions and comments:
1. The authors mentioned "Weights of the feature encoder are not updated at fine-tuning time". Is there a specific reason? Is it because after updating the encoder parameters the performance gets worse?
2. A follow-up. "The purpose of this analysis is not to recover full language families but to better understand how our model allocates the latent representations across languages. ". It could be very interesting if some connections can be found between phonological similarities among languages (https://wals.info/feature) and their latent speech representations. For example, how come the black languages (Chinese) are closer to purple languages than blue languages in figure 2 (b)
3.  Wondering why the authors don't use tsne for visualization.

---

### Official Review · AnonReviewer2 · 2020-10-28
**A good multilingual system paper for speech recognition, but more on application side**

**Rating:** 4
**Confidence:** 5

**Review:**

This paper applied self-supervised learning (wav2vec 2.0) to multilingual speech recognition task. Promising results demonstrated in Babel and Mozilla tasks.

All the key points studied in this paper has being shown in previous paper, e.g. wav2vec 2.0 shows the effectiveness of self-supervise learning to speech, a long history of speech papers shows multilingual pretrain is good for mono lingual, on low resource task for both e2e or hybrid system. Combining all of these are new, but this is more like a system paper and fit better to a speech conference.

Section 4.4.2 is interesting, but given https://arxiv.org/pdf/1901.08810.pdf has similar analysis (in their case, quantized unit related to phonemes), this analysis method also has being studied. Moreover, applying latent output from a multilingual ASR system to language identification task has being studied before and works very well.

I'd assume the paper give very good results. however, it ignores almost all previous publications on Babel, e.g. there is no comparison with a hybrid model. Table 2 is self-contained, but it would be great also include some previous Babel number so we can see how e2e compare with traditional hybrid model.

This is a well written and interesting paper. However, there isn't much add to wav2vec 2 and multilingual (transfer) learning methodology. As a milestone paper, for example, super good performance, these task are not representative enough. So I don't think it fit to ICLR and suggest it resubmit to a speech conference.

---

### Official Review · AnonReviewer3 · 2020-10-28
**Very good phone recognition results on BABEL task**

**Rating:** 6
**Confidence:** 3

**Review:**

Authors extended the XLSR model from the previous mono-lingual task, where self-supervised learning was used in representation learning, to multi-lingual task. The basic idea does make a lot of sense, where all the data, irrespective of the language, is pushed through the representation learning task.  We are anyways talking about human speech, so it is reasonable to assume that learning one language can help in learning the other language.

My main criticism is that technical novelty appears to be limited. The core idea appears to be just the use of multi-lingual data instead of mono-lingual data. Authors could emphasize the technical novelty in the Introduction if I am not correct in this assessment. On the other hand, the results authors obtain are excellent so there is merit in publishing them.

Other issue I have is that BABEL has been used in speech community quite a bit, so I am not completely sure whether authors baseline results are the current SoTA. Especially, I find a bit dubious that  (Fer et al., 2017) is used as a baseline, where that paper is dated and deals with language identification and not multi-lingual phone recognition! Actually, these results are not even available from (Fer et al., 2017). In speech community, WER results are typically reported and not PERs, this might make comparison to SoTA a bit more difficult as authors do not seem to use language model at all. So apples-to-apples comparison might be harder. However, I ask authors to make sure the baselines are good enough.

The other thing is that unsupervised ASR has seen wealth of papers recently, some have used GAN type approaches such as https://arxiv.org/abs/1904.04100 authors could also comment a bit about these approaches. Note in that one PER is also used as a measurement.

---

### Official Review · AnonReviewer1 · 2020-10-28
**Very informative results of using a semi-(self-) supervised learning for multilingual ASR**

**Rating:** 5
**Confidence:** 5

**Review:**

This paper proposes to apply a state-of-the-art self-supervised training method (wav2vec 2.0) to multilingual ASR. The paper provides an intensive analysis of how we use such pre-training-fine-tuning strategies, including 1) multilingual data balance, 2) language-independent pre-trained/fine-tuning model vs. language-dependent pre-trained/fine-tuning model, use of a large model and 3) the visualization of the embedding of each language.  The paper also has some comparisons with the other reports in the same setup. This multilingual ASR scenario is one of the most straightforward applications of self-supervised training, and this paper has significant contributions to the ASR research communities. However, if we consider this paper's contributions in more broad machine learning perspectives (main target in ICLR), they are not significant. First, the method does not have the formulation or algorithm level distinction from the original wav2vec 2.0 paper. The application of this paper is also limited to ASR. The paper should address more applications. I think that we can apply the same methodology to speech translation. Such kinds of applications make the paper better shape in terms of the generalization of the proposed multilingual self-supervised training methodology.

Additional comments
- I suggest the authors to consider to submit this paper to ICASSP or Interspeech. The paper would gain more attention in these conferences.
- There are several complicated tweaking, and I'm concerned about the reproducibility. It would be great if the authors release the recipe for this study.
- I could not find the detail about the decoding information, e.g., with or without beam search and language models.
- I'm curious about the robustness (sensitivity) of the noise, recording condition, and speaking styles. For example, what happens if we mix both commonvoice and Babel data since their recording conditions and speaking styles are different. I'd like to have some analysis and visualization results.
- Sections 2.1 and 2.2 are hard to follow, and they need more elaborated explanations. For example, this part is not very self-consistent, and we need to check the original wav2vec 2.0 paper to understand it (even high-level). No explicit definition about $T$, $G$, $V$, and $p$.
- Section 3: It is not easy to follow wow to use the data for each condition. I suggest the authors summarize such information by using a table.
- Many experimental results are very informative, but for me, the most informative result is that if we use a large model, we could use a single multilingual model. I think this finding would be beneficial for our practical multilingual ASR development.

---

### Decision · Program_Chairs · 2021-01-07
**Final Decision**

**Decision:**

Reject

**Comment:**

This work mainly applies wav2vec 2.0 to multilingual speech recognition and lacks of novelty.
The various pre-training and fine-tuning mix-match are specific to the speech recognition task. As suggested by reviewers, it is recommended to resubmit to a speech conference.
Also the paper lacks comparisons to SOTA on one of the well studied task (i.e. BABEL) in the speech field.

The main factor for the decision is lack of novelty.